


# Prediction and variation of auroral oval boundary based on deep learning model and space physical parameters

Yiyuan Han [1], Bing Han[1], Zejun Hu[2], Xinbo Gao[1], Lixia Zhang[1], Huigen Yang[2], Bin Li[2]

[1]School of Electronic Engineering, Xidian University, Xi'an 710071, China
[2]SOA Key Laboratory for Polar Science, Polar Research Institute of China, Shanghai 200136, China

*Correspondence to*: Bing Han (bhan@xidian.edu.cn)

**Abstract.** The auroral oval boundary represents important physical process with implications for the ionosphere and magnetosphere. An automatic auroral oval boundary prediction method based on deep learning in this paper are applied to study the variation of auroral oval boundary, associated with different space physical parameters. We construct an auroral oval
boundary dataset to train our proposed model, which consists of 184416 auroral oval boundary points extracted from 3842 UVI images captured by Ultraviolet Imager of the Polar satellite and its corresponding 18 space physical parameters selected from OMNI dataset during December 1996 to March 1997. Furthermore, several statistical experiments and correlation analysis experiment are performed based on our dataset to explore the relationship between space physical parameters and the location of auroral oval boundary. The experiment results show that the prediction model based on deep learning method could
estimate auroral oval boundary efficiently, and different space physical parameters have different effects on auroral oval boundary, especially interplanetary magnetic field (IMF), geomagnetic indexes and solar wind parameters.

## 1 Introduction

Auroral oval is a circular belt of auroral emission around magnetic poles (Loomis, 1890; Akasofu, 1964). The auroral oval poleward and equatorward boundaries are related to geophysical parameters, which can implicit for the coupling process
between the solar wind, ionosphere and magnetosphere. For example, the polar cap ionosphere, which is considered as an area of opening magnetic field inside auroral oval poleward boundary. This area is closed related with energetic particle entrance from heliosphere to earth's atmosphere. So, the segmentation and prediction for auroral oval boundary is very significant for studying on certain physical events.

In the past few decades, scholars have constructed ex-tensive researches on the relationship between location of auroral
oval boundary and space physical parameters (Niu et al., 2015). Feldstein proposed that the position of auroral oval boundary is correlated with the Q-index of magnetic activity on the nightside of earth (Feldstein and Starkov, 1967). Starkov and Holzworth expressed that inner and outer boundaries of auroral oval can change with geomagnetic indexes and IMF (Holzworth and Meng, 1975; Holzworth and Meng, 1984; Starkov, 1994(a)) Starkov designed some simple formulas on the location of auroral oval and diffuse aurora. Variations of the size of polar cap, auroral oval and diffuse aurora are regarded as





three independent function variables of AL index (Starkov, 1994(b)). Carbary constructed a-Kp-related model of auroral oval boundary by binning UVI images from different months (Carbary, 2005). For describing the particle precipitation characteristics, Zhang proposed a-Kp-dependent model of the mean energy and energy flux precipitating electrons in auroral oval (Zhang and Paxton, 2008). Sigernes com-pared the methods which proposed by Zhang and Starkov to calculate the size

and position of auroral oval using a Kp-based function. (Sigernes et al., 2011). Milan proposed a model based on average proton and electron of auroral images from three years observed by the IMAGE spacecraft. The experiment demonstrated that Kp, solar wind parameters including so-lar wind velocity, density, and pressure, interplanetary magnetic field (IMF) magnitude and orientation have effect on the intensity and shape of auroral oval (Milan, 2010). Hu and Yang used the segmentation results of auroral oval obtained from UVI on Polar satellite to build connection be-tween the position of auroral oval boundary and

AE index, IMF and solar wind parameters by using multiple regression method (Hu et al., 2017; Yang et al., 2016). Ding presented a C-means clustering algorithm based on fuzzy local information to extract auroral oval poleward and equatorward boundaries from merged images with filled gaps captured from both GUVI and SSUSI (Ding et al., 2017). However, the position of auroral oval boundary is not determined by one space physical parameter, those methods mentioned above just only used one or several space physical parameters to explore the relationship be-tween space physical parameters and auroral

oval boundary. We can't determine whether other space physical parameters can influence the location or size of auroral oval. And we also don't know whether the mapping relationship between space physical parameters and auroral oval boundary is linear or nonlinear.

     As we know, machine learning has been applied to many fields, including medical, traffic, space physics and other interdisciplinary fields. Recently, deep learning models have led to a series of breakthroughs on image classification, object

detection, image recognition and other fields. Conventional machine learning methods have some limitations for processing complex data, especially in space physics field. There are no suitable internal features, such as shape, colour and so on. Therefore, many effective machine learning methods can't obtain satisfied performance on pro-cessing space physics data. While, deep learning methods are representation-learning methods with multiple levels of representation. It has turned out to be very good at discovering intricate structures in high dimensional data and multimodal data (LeCun et al., 2015).

In this paper, a new automatic auroral oval boundary prediction model is proposed based on deep learning method. The experiment results show that the model pro-posed in this paper can predict aurora oval boundary accurately by using space physical parameters and the location of auroral oval boundary at the previous moment. In addition, we explore the effect of every space physical parameter on auroral oval boundary. The rest of this paper is organized as follows. Sect. 2 describes our proposed algorithm to predict auroral oval boundary in detail. The experiment analysis and discussion are given in Sect. 3,

including dataset construction, subjective and objective evaluation, the selection of model parameters and the discussion about influence of every space physical parameter on auroral oval boundary. Finally, we draw several conclusions in Sect. 4.





## 2 Prediction of auroral oval boundary based on deep learning method

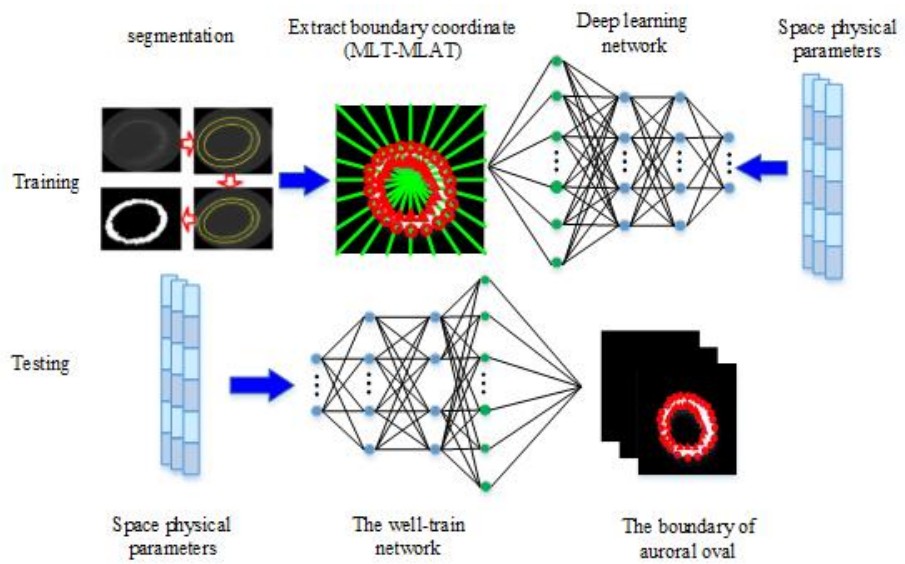

**Figure 1: The flowchart of auroral oval boundary prediction model based on deep learning.**

The flowchart of auroral oval boundary prediction model is shown in Fig. 1. There are two major steps in our pro-posed model,

pre-training on our dataset and online prediction. In the training phase, auroral oval images are usually affected by heavy noise and other interference. So, the auroral oval boundary is blurred and it is difficult to find from background. Compared with other image segmentation methods, MRSM (Liu et al., 2013) can eliminate the cumbersome process of adjusting parameters and has better segmentation accuracy. We use MRSM firstly to extract positions of auroral oval boundary. The center of auroral oval spatial distribution in magnetic local time-magnetic latitude coordinate is located in the geomagnetic pole. The magnetic

latitude of auroral oval usually ranged from 57.5 degree to 73.5 degree according to the statistic studies on previous work (King and Papitashvili, 2014). In order to unify the distribution of aurora oval boundary, the coordinates of those extracted boundary points are transformed into MLT-MLAT coordinate secondly. Finally, these transformed boundary points and its corresponding space physical parameters were input into deep learning network to train our prediction model. In the testing phase, we can obtain the corresponding boundary points of auroral oval by sending those space physical parameters and the

position of auroral oval boundary points at the previous moment to our well-trained network.

The deep learning network is constructed by a two-layer Restrict Boltzmann Machine (RBM) network (Hinton et al., 2006; Yu and Deng, 2011) and a Radial Basis Function (RBF) network (Łukaszyk, 2004). The computational processing of RBM and RBF is illustrated by Eq. (1)-(4). In the training phase, the input of RBM network are 18 space physical parameters from OMNI dataset and coordinate values of auroral oval poleward and equatorward boundaries extracted from segmented UVI images with MRSM. It can be represented as $X = [x_1, x_2, \cdots, x_m]^T$, where $m$ is the number of network nodes. The first

layer of RBM network is denoted as $\theta_1 = \{w_{i_1 j_1}, a_{i_1}, b_{j_1}\}$, where $w_{i_1 j_1}$ is the weight between the visible unit $i_1$ and the hidden





unit $j_1$, $a_{i_1}$ is the bias of visible unit $i_1$, $b_{j_1}$ is the bias of hidden unit $j_1$. The hidden layer of the first layer in RBM network is the visible layer of the second layer in RBM network, which is denoted as $\theta_2 = \{w_{j_1 j_2}, a_{j_1}, b_{j_2}\}$, where $w_{j_1 j_2}$ is weight between the visible unit $j_1$ and the hidden unit $j_2$, $a_{j_1}$ is the bias of visible unit $j_1$, $b_{j_2}$ is the bias of hidden unit $j_2$. The output of the first layer of RBM network is denoted as $Y_1 = [y_{11}, y_{12}, \cdots, y_{1n}]^T$, where $n$ denotes the nodes number of the first layer in RBM

network.

$$y_{1 j_1} = \sum_{i=1}^{m} x_{i_1} w_{i_1 j_1} + b_{j_1} \quad j_1 = 1,2,\cdots,n \tag{1}$$

The output of the second layer of RBM network is denoted as $Y_2 = [y_{21}, y_{22}, \cdots, y_{2c}]^T$, where $c$ denotes the nodes number of the second layer in RBM network.

$$y_{2 j_2} = \sum_{j=1}^{n} x_{j_1} w_{j_1 j_2} + b_{j_2} \quad j_2 = 1,2,\cdots,c \tag{2}$$

Finally, since Contrastive Divergence (CD) (Hinton, 2002) is an approximation of the log-likelihood gradient what has been found to be a successful update rule for training RBM, we can obtain a well-trained RBM network by CD.

The function of RBF network can make the output of RBM network infinitely approximate to the coordinate values of auroral oval boundary by a radial basis function. The input of RBF network is the output of second layer in RBM network. The output of RBF network is represented as $Y = [y_1, y_2, \cdots, y_d]^T$, where $d$ denotes the number of the output layer nodes. $w_{j_3 o}$ is weight

between hidden unit $j_3$ and the output node $o$. $l$ is the number of radial basis function. $\varphi_{j_3}$ is the $j_{th}$ radius basis function and $c_{j_3}$ is the center of $j_{th}$ radial basis function. $\sigma_{j_3}$ is the center width of radial basis function.

$$y_o = \sum_{j_3=1}^{l} w_{j_3 o} \varphi_{j_3}(||Y_2 - c_{j_3}||) \quad o = 1,2,\cdots,d \tag{3}$$

$$\varphi_{j_3}(||Y_2 - c_{j_3}||) = exp(-\frac{||Y_2 - c_{j_3}||^2}{\sigma_{j_3}^2}) \quad j_3 = 1,2,\cdots,l \tag{4}$$

## 3 Experiments and results analysis

### 3.1 Dataset construction and evaluation criterion

The auroral oval images used in this paper are captured by Ultraviolet Imager (UVI) which is a 2-D snapshot type camera on Polar satellite. The UVI on Polar satellite has acquired more than several millions of images during its entire mission. In April 2008, it no longer works. There was no effective observation after 2000, because Polar satellite changed its view after 2000. In order to balance the relationship between spatial resolution and global coverage, the spatial resolution of UVI is 30KM at

apogee, the CCD array on board has 224*220 pixels and the single pixel spatial resolution is 0.0036 degree and 0.04 degree in two directions respectively. The unilluminated edges of CCD are discarded, which results in the frame size of an auroral oval image is 200 by 228 pixels, and the frame rate is 37s. There are 4 band sensors aboard satellite usually, UVI images in our dataset are derived from Lyman-Brige-Hopfield long band (160-180mm). In our experiments, each auroral oval image is divided into 24 magnetic regions centered on geomagnetic pole according to magnetic local time (MLT). As shown in Fig. 2,





the intersection points between auroral oval boundary and division line are extracted. 48 boundary points are gathered from one auroral oval image. The poleward and equatorward boundary points are marked as red triangle and red circle respectively.

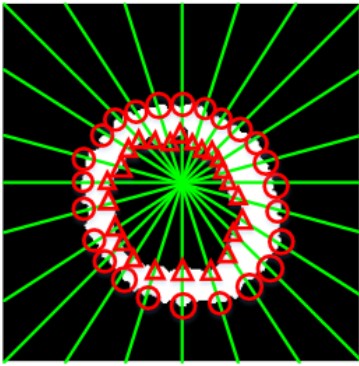

**Figure 2: The schematic of extracting auroral boundary points.**

5 **Table 1: Space physical parameters selected from OMNI dataset.**

| Parameter name | Units |
|---|---|
| Bx | nT |
| By | nT |
| Bz | nT |
| Flow Speed (Vp) | Km s$^{-1}$ |
| Proton density (Np) | n cc$^{-1}$ |
| Temperature | K |
| Flow pressure (pdyn) | nPa |
| Electric Field | Mv m$^{-1}$ |
| Plasma beta | - |
| Alfven mach number | - |
| AE-1-min AE-index | - |
| AL-1-min AL-index | - |
| AU-1-min AU-index | - |
| SYM/D-1-minute SYM/D index | - |
| SYM/H-1-minute SYM/H index | - |
| ASY/D-1-minute ASY/D index | - |
| ASY/H-1-minute ASY/H index | - |
| PC-1-minute Polar Cap index | - |

The space physical parameters downloaded from NASA OMNI dataset with different time resolution. It is common knowledge that IMF, solar wind parameters, geomagnetic Indexes have a time resolution of 1 min, the other space physical parameters





maybe have a higher time resolution. According to the effect from other circumstance factors, such as, the time to traverse magnetosphere and Alfven wave, not all the response time of auroral events are equal to its propagation time. We align the time of all space physical parameters with the time of UVI images in our dataset to avoid the problem of different time resolution between space physical parameters and auroral oval images. In OMNI dataset, we selected 18 space physical

parameters including the common parameters which has been verified to be related to the position of auroral oval boundary (Holzworth and Meng, 1975; Starkov, 1994(a); Starkov, 1994(b); Milan et al., 2010; Hu et al., 2017) and some unfamiliar parameters which are never discussed in previous works. Therefore, our dataset includes 184416 auroral oval boundary points extracted from 3842 UVI images and its corresponding values of 18 space physical parameters. Table 1 shows 18 space physical parameters we used in this paper.

In order to evaluate the precision of predicted auroral oval boundary points by our model, we use the common metric MAE (Mean Absolute Error) to assess the error between predicted auroral oval boundary points and real auroral oval boundary points. The MAE (Mean Absolute Error) can be defined as Eq. (5).

$$MAE = \frac{1}{24} \sum_{i=1}^{24} \left( \frac{1}{k} \sum_{j=1}^{k} |F_{MLAT}^{ij} - S_{MLAT}^{ij}| \right) \tag{5}$$

$S_{MLAT}^{ij}$ represents MLAT of the $j_{th}$ test sample at $j_{th}$ MLT region obtained from the segmented image, and $F_{MLAT}^{ij}$ indicates

MLAT of the $j_{th}$ test sample at $i_{th}$ MLT region acquired by our prediction model. $k$ is the total number of test samples.

### 3.2 Parameters setup of deep learning network

Since the effectiveness of prediction model is influenced by the number of hidden layer nodes in RBM network (Hinton, 2012) and the training error of RBF network, we build two experiments to find the most suitable parameters for our network. For both experiments, space physical parameters and position of poleward and equatorward boundary points in 24 MLT regions

of 3000 UVI images are selected as training samples, the remaining are regarded as test samples. In experiment 1, the training error of RBF network is set to 4 magnetic latitude and the number of hidden layer nodes in RBM network are 32, 64, 96 and 128 respectively. We use the average MAE with 100 experiments to verify the stability of our model, because training samples and test samples were divided by random number. The corresponding MAE is shown in Fig. 3(a). From the Fig. 3(a), MAE reaches the smallest value when the number of hidden layer nodes are set to 32. In experiment 2, the number of hidden layer

nodes are set to 32 according to the results in experiment 1. There often has overfitting problem when we train a neural network (Krizhevsky et al., 2012). Overfitting can be interpreted as a phenome-non, which is the model performs well on training set and unsatisfactorily on test set. We set different training error to avoid overfitting problem. So, the training error of RBF network is set to 2, 4, 6 and 8 magnetic latitudes empirically. The corresponding MAE is shown in Fig. 3(b), and MAE reaches the minimum when the training error of RBF net-work is 4 magnetic latitudes. From the two experiment results above, we set

the number of hidden layer nodes in RBM network and the training error of RBF network to 32 and 4 respectively as the optimal parameters of deep learning network in the following experiments.



To demonstrate the availability of our proposed model, we compare the proposed model with Back Propagation (BP) network (Rumelhart, 1986) and Yang's model (Yang et al., 2016). The subjective prediction results obtained by the three method are shown in Fig. 4, circles and squares stand for poleward boundary points and equatorward boundary points which are obtained from the segmented image, '＋' and '×' marks represent poleward boundary points and equatorward boundary points which

are obtained from our prediction model. Although these three methods have similar prediction results in most areas on auroral oval boundary, it is obviously that our method can obtain the more accurate boundaries than the other two compared methods, where marked by blue rectangle and red rectangle in Figure 4. In more detail, the results of BP model and our model are shown in Fig. 4(a) and Fig. 4(b) respectively, we can clearly see that the distances between auroral oval boundary points predicted by our method and real auroral oval boundary points are smaller than the distances between BP model's results and real auroral

oval boundary points in red rectangle areas. From Fig. 4(a) and Fig. 4(c), our pre-diction points are closer to real auroral oval boundary points compared with Yang's prediction points in blue rec-tangle area. Meanwhile, the MAE of different methods are shown in Table 2. From this table, our method has the smallest MAE not only in poleward boundary bur also in equatorward boundary, because our model can extract more useful information and feature from auroral oval images than the other two models. As a consequence, we can draw the conclusion that the proposed model in this paper is more suitable for predicting

auroral oval boundary.

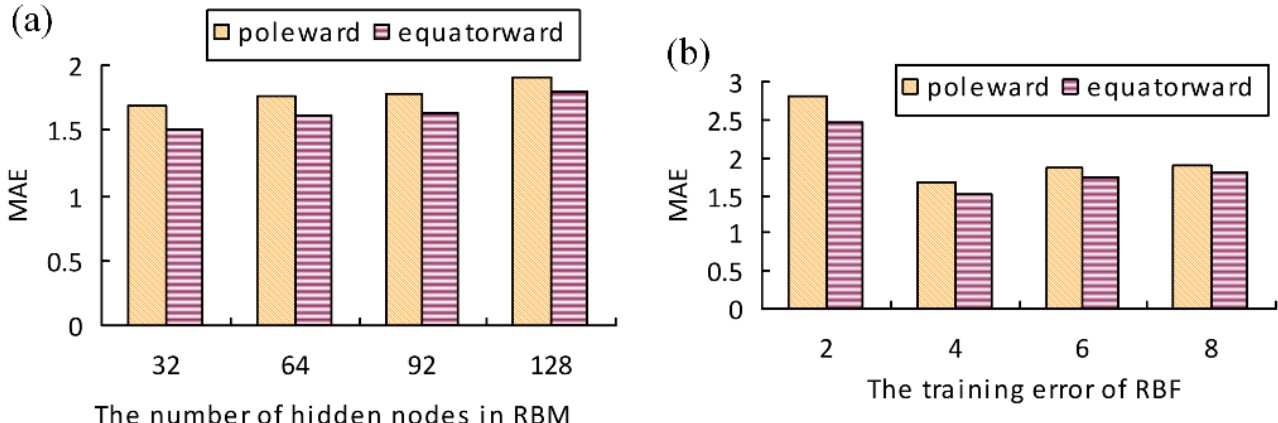

**Figure 3: (a)The MAE value of different hidden layer node, (b) The MAE value of different training error.**

**Table 2: The MAE value of different methods.**

| methods | BP | Yang's | ours |
| --- | --- | --- | --- |
| Poleward boundary | 2.20 | 2.01 | 1.69 |
| Equatorward boundary | 2.19 | 1.91 | 1.51 |


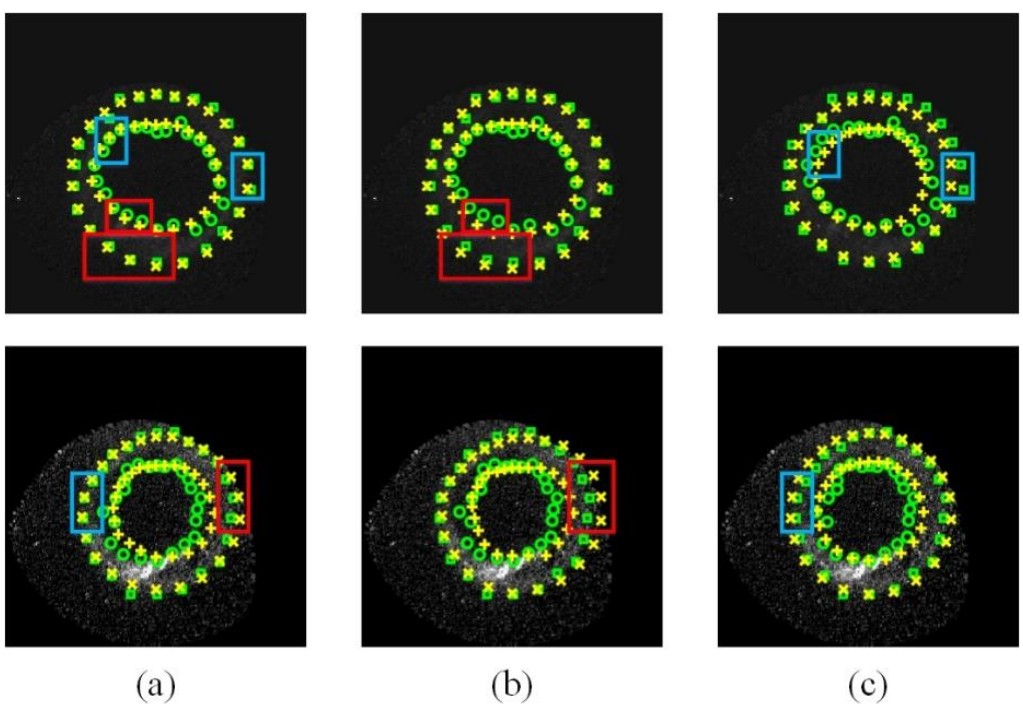

**Figure 4: The different subjective results based on different methods. (a) The subjective results predicted by our method (b) The subjective results predicted by BP network (c) The subjective results predicted by Yang's method**

### 3.3 The influence of space physical parameters on auroral oval boundary

As we known, the location of auroral oval boundary is affected by a variety of space physical parameters. Variation of auroral oval boundary in different MLT sectors are related to different space physical parameters. For sake of exploring the influence of space physical parameters on poleward and equatorward boundaries specifically, the boundary points are further processed as follows (Hu et al., 2017). Firstly, all poleward and equatorward boundary points are divided into 24 subsets of poleward and equatorward boundary points according to 24 MLT sectors. Secondly, in every MLT subset, we sort boundary data with

respect to the value of all space physical parameters, and divide evenly boundary data into 10 groups. In order to observe the variation tendency of each parameter in different MLT sectors clearly, in every MLT sectors, the relationship be-tween each space physical parameter and the location of auroral oval boundary was represented as a Quadratic Equation based on the principles of the least square conic fitting (Fitzgibbon et al., 1999). Then, we calculate the location of poleward and equatorward boundary points for each space physical parameter using this function. Finally, we use the boundary data which

calculated by Quadratic Equation to discuss the influence of space physical parameters on auroral oval boundary. In this section, we build 3 statistical experiments to discuss how IMF, solar wind parameters and geomagnetic indexes influence on auroral oval boundary. An auroral oval boundary prediction experiment by inputting every single space physical parameter to explore the relationship between auroral oval boundary and 18 space physical parameters. And a correlation analysis experiment is con-structed to study the connection between combination of different space physical parameters and auroral oval boundary.

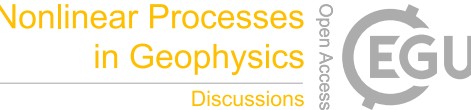

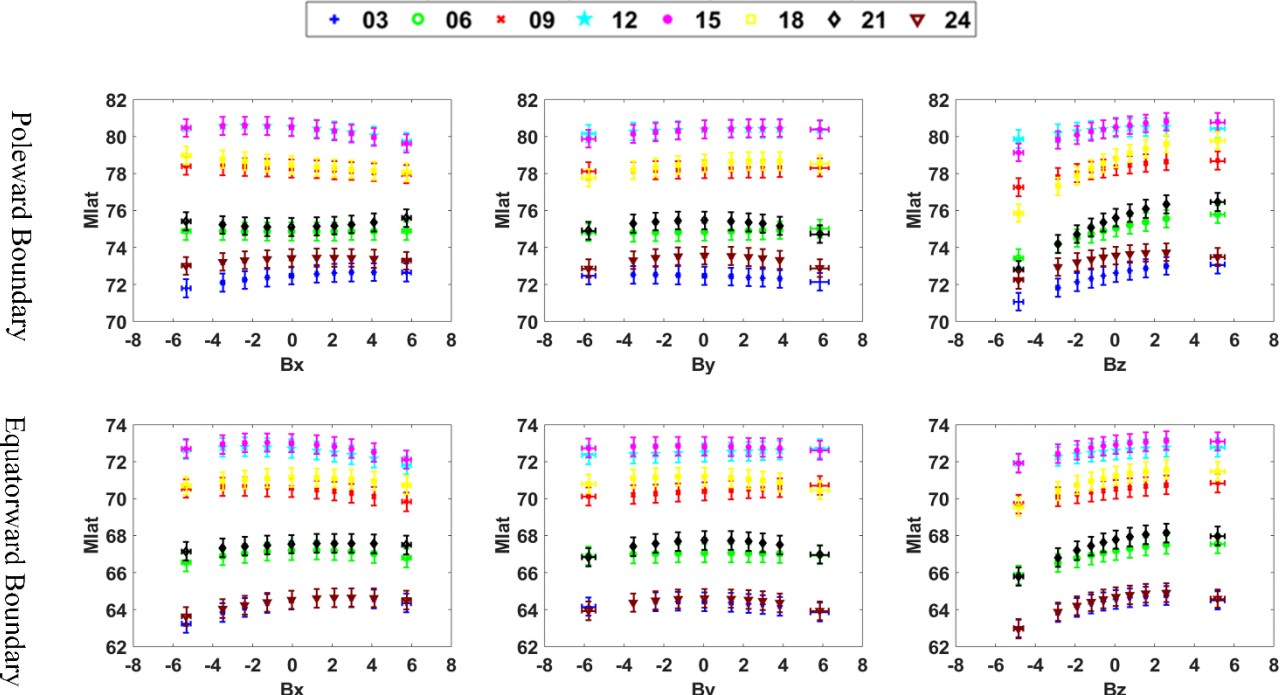

**Figure 5: Response of magnetic latitude of poleward (left column) and equa-torward (right column) boundaries to Bx, By and Bz respectively at 0030,0060,0090,1200,1500,1800,2100 and 2400 MLT.**

### 3.3.1 Experiment 1: Influence of different IMF components on Aurora oval boundary

The IMF can affect auroral oval boundary through different space processes. In this experiment, response of different IMF components to auroral oval boundary are shown in Fig. 5. The different colour and shape markers represent different MLT sector. The vertical error bars represent one eighth of standard deviation from mean value of auroral oval boundary position, and the horizontal error bars rep-resent standard deviation from mean value of different IMF components in each binned data. Therefore, the length of vertical error bar is fixed, the length of the horizontal error bars is changeable because of different

standard deviation in each binned data.

From Fig.5, we can see that the poleward and equatorward boundaries in each MLT sector show a step-by-step poleward displacement with the increase of IMF Bz component. It has been widely accepted that IMF Bz controls the energy coupling between the solar wind and the magnetosphere (Cho et al., 2010; Makita et al., 1983). During a period of southward IMF (Bz < 0), poleward motion of auroral oval boundary is due to a higher reconnection rate in the process of dayside reconnection.

However, most poleward motion of auroral oval boundary occurred during northward IMF (Bz > 0). Under northward IMF (Bz > 0) condition, poleward activity of auroral oval boundary often related to IMF By component (Xing et al., 2013). The poleward and equatorward boundaries in 09:00-15:00 MLT show a gradually poleward displacement with the rise of IMF By component, and the poleward and equatorward boundaries in 18:00-06:00 MLT gradually approaches to pole with the decrease of absolute of IMF By component from Fig.5. This statistical discovery proves previous studies on IMF By component. Such

as, Karlson's observations suggests that IMF By component is related to prenoon-postnoon asymmetry of poleward activity (Karlson et al., 1996). And it is well known that ionospheric convection is mainly controlled by IMF Bz and By components (Cowley and Lockwood, 1992; Huang et al., 2000), which implies the prenoon-postnoon asymmetry of pole-ward activity is similar to the procedure of ionospheric plasma convection. Both two space activities mentioned above are affected by the

variety of IMF By component (Provan et al., 1999). The poleward and equatorward boundaries in 21:00-06:00 MLT show a gradually poleward motion with the ascent of IMF Bx component observed from Fig.5, which is consistent with IMF By and Bz.

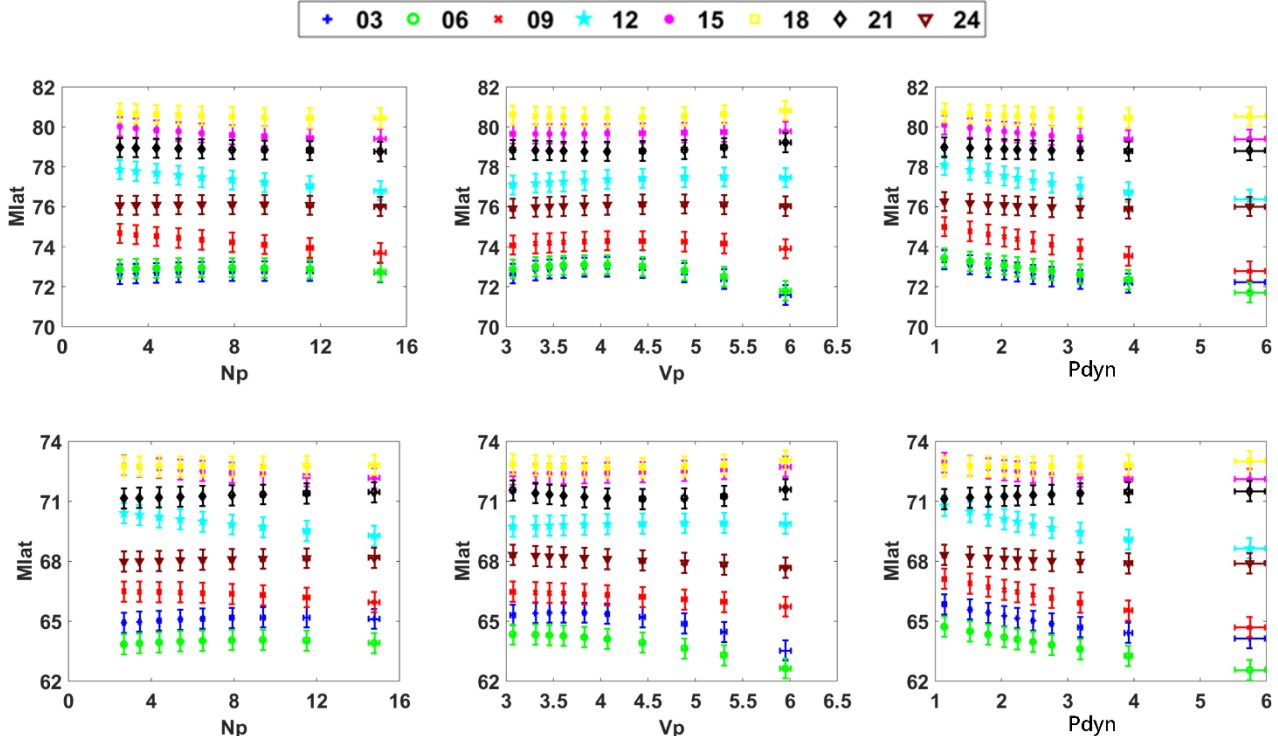

**Figure 6: Response of magnetic latitude of poleward (left column) and equa-torward (right column) boundaries to Np, Pdyn and**
**Vp respectively at 0030,0060,0090,1200,1500,1800,2100 and 2400 MLT.**

### 3.3.2 Experiment 2: Influence of different solar wind parameters on Aurora oval boundary

For sake of finding the variation trend of auroral oval boundary with the change of solar wind parameters, including solar wind density (Np), solar wind speed (Vp) and solar wind dynamic pressure (Pdyn) respectively, experiment 2 is performed. Fig.6 shows the response of different solar wind parameters on auroral oval boundary.

From Fig.6, both poleward boundary and equa-torward boundaries shrink in 21:00-06:00 MLT when the value of Np rises. Meanwhile, poleward and equatorward boundaries in 09:00-18:00 MLT gradually approaches equator when the value of Np rises. Besides, we can obtain the following conclusions: The poleward and equatorward boundaries in 03:00-18:00 MLT expand to equator clearly with the increase of Pdyn. And equatorward boundary in 21:00-24:00 MLT has a poleward motion



with the increase of Pdyn. There has an obvious poleward motion in nightside sector impacted by the increscent Pdyn and Np according the conclusions above. We can draw a coincident inference with previous studies. For examples, poleward displacement of auroral oval boundary along with the increscent Pdyn, which results from the shrunken polar cap (Cho et al., 2010). By extension, there must have some dependencies between the varying size of polar cap and nightside reconnection

(Boudouridis et al., 2003). Compared with the change of auroral oval boundary in nightside sector, both poleward and equatorward boundaries are enlarged when the value of Pdyn and Np rise, which observed from Fig.6. Previous explorations and simulations shown that enlarged Pdyn can enhance ionospheric potential and the corresponding field-direction current intensity, which can lead to increasement of global auroral activity intensity. Meanwhile, the position of auroral oval boundary will ex-tend to low latitudes (Peng et al., 2011). From Fig.6, it appears a distinct equatorward movement with increase of Vp

in 24:00-06:00 MLT for both poleward boundary and equatorward boundary. This changing pattern of Vp and auroral oval boundary which we illustrate above is consistent with Hu's study in 2017 (Hu et al., 2017).

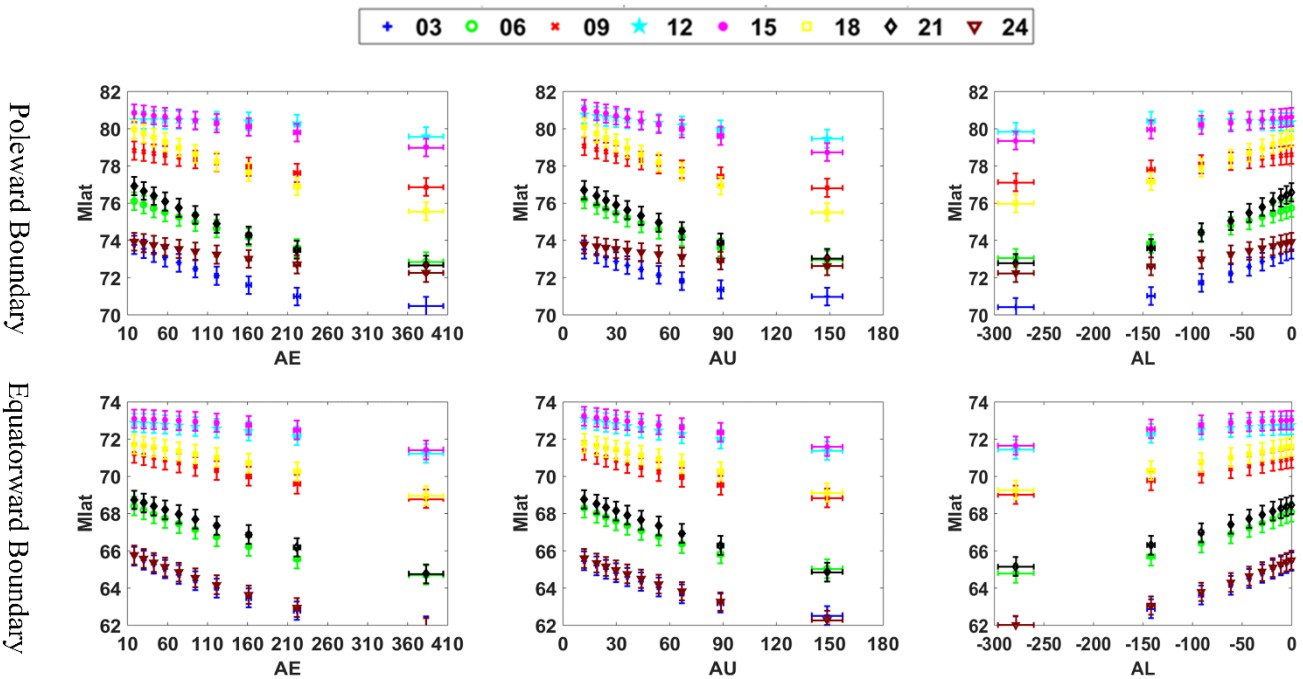

**Figure 7: Response of magnetic latitude of poleward (left column) and equa-torward (right column) boundaries to AE, AU and AL respectively at 0030,0060,0090,1200,1500,1800,2100 and 2400 MLT.**

**3.3.3 Experiment 3: Influence of geomagnetic indexes on Aurora oval boundary**

In this experiment, the average tendency of poleward and equatorward boundaries influenced by geomagnetic index-es (AE, AL, AU) shows in Fig.7.

As we can see from Fig.7, in every MLT sector, pole-ward and equatorward boundary move to low magnetic latitude with the ascending AE and AU index. While, pole-ward and equatorward expand to high magnetic latitude with the ascending





of AL index. AE index is often used to characterize the strength of substorm activity in magneto-sphere. Therefore, it can be considered that auroral oval expands to equator due to the enhancive substorm activity. Furthermore, the amount of energy enters magnetotail along with the strengthening of substorm activity. It means that AE index will increase when energy in magnetotail re-leased through substorm, which is coincident with our found about AE from Fig.7.

5  **Table 3: The MAE influenced by different space physical parameters.**

| Parameter name | MAE(poleward/equatorward) |
|---|---|
| Bx | **1.6222/1.4448** |
| By | 1.6134/1.4462 |
| Bz | 1.6139/1.4476 |
| Flow Speed (Vp) | 1.6117/1.4485 |
| Proton density (Np) | **1.6285/1.4451** |
| Temperature | 1.6129/1.4458 |
| Flow pressure (pdyn) | **1.6242/1.4463** |
| Electric Field | 1.6113/1.4430 |
| Plasma beta | 1.6118/1.4435 |
| Alfven mach number | 1.6193/1.4473 |
| AE-1-min AE-index | 1.6183/1.4562 |
| AL-1-min AL-index | **1.6325/1.4668** |
| AU-1-min AU-index | 1.6117/1.4517 |
| SYM/D-1-minute SYM/D index | 1.6187/1.4500 |
| SYM/H-1-minute SYM/H index | 1.6197/1.4581 |
| ASY/D-1-minute ASY/D index | 1.6137/1.4550 |
| ASY/H-1-minute ASY/H index | 1.6079/1.4500 |
| PC-1-minute Polar Cap index | 1.6120/1.4512 |

### 3.3.4 Experiment 4: Influence of all 18 space physical parameters on Aurora oval boundary

As we know, most of studies on how the space physical parameters affect auroral oval boundary are focus on solar wind parameters, geomagnetic indexes and IMF components. There are lots of corresponding conclusions about the influence of those space physical parameters on auroral oval boundary up to now. Nonetheless, how the other space physical parameters not mentioned above affect the auroral oval location has not been addressed. In order to further explore the variation of auroral oval boundary influenced by different space physical parameters, the experiment 4 is performed. In experiment 4, we send one physical parameter selected from Table 1 at the present moment and the coordinates of auroral oval boundary points at the previous moment to our prediction model. And the output of our model are 48 coordinates values of auroral oval boundary





points and the MAE between real boundaries and predicted boundaries. The MAE of poleward and equatorward boundaries influenced by different space physical parameters are given in Table 3. We can infer the response of auroral oval boundary to 18 space physical parameters through the different MAE of these space physical parameters.

The MAE of boundary position is 1.6076 and 1.4545 respectively when we only use boundary positions at the previous

moment to predict poleward and equatorward boundaries. We take this MAE as standard, called S-MAE. Compared with the S-MAE, we can see that the MAE in-crease about 1.9% for poleward boundary by adding any-one space physical parameters into input of our model from Table 3. Meanwhile, the variety of MAE for equa-torward boundary is between -0.7% and 0.7%. Although different space physical parameters have different influ-ences on auroral oval boundary, compared to other space physical parameters in Table 3, the MAE of auroral oval boundary can display the greatest impact when AL, Bx, Np, Pdyn are

used as the input of our model respectively, which suggests that these 4 space physical parameters mentioned above have a great influence on the position of auroral oval boundary.

**Table 4: The Pearson correlation coefficient of all 18 space physical parameters from Dec. 1996 to Mar. 1997.**

| Parameter name | Correlation coefficient |
|---|---|
| Vp-Np | -0.5970 |
| Vp-SYM/H | -0.5120 |
| Np-Pdyn | 0.7662 |
| Np-SYM/H | 0.5584 |
| AE-AL | -0.9437 |
| AE-AU | 0.7139 |
| AE-PC | 0.8067 |
| AL-PC | -0.7079 |
| AU-PC | 0.6924 |

**3.3.5 Experiment 5: Correlation analysis of all 18 space physical parameters**

For the sake of analyzing the influence of space physical parameters on auroral oval efficiently, we not only consider the effect

of each space physical parameter on auroral oval boundary, but also take the effect on auroral oval boundary with different combinations of space physical parameters into account in experiment 5. As a result, we first calculate the correlation of all 18 space physical parameters using Pearson correlation coefficient, which is a statistic value that reflects the degree of linear correlation between two variables. The Pearson correlation coefficient of two variables $(X, Y)$ equals the covariance of the two variables $(X, Y)$ divided by the product of their standard deviations $(\sigma_X \sigma_Y)$. The formula of Pearson correlation coefficient is

can presented as Eq. (6), and the Pearson correlation coefficient of all 18 space physical parameters are given in Table 4. The process of experiment 5 is similar to experiment 4 and the MAE of different space physical parameter combinations are given in Table 5.





$$\rho_{X,Y} = \frac{cov(X,Y)}{\sigma_X \sigma_Y} = \frac{E[(X-\mu_X)(Y-\mu_Y)]}{\sigma_X \sigma_Y} \tag{5}$$

The MAE of poleward and equatorward boundaries by using three components of IMF and auroral oval boundary positions at the pervious moment as the input of proposed model are 1.6313 and 1.4759 severally in Table 5. The MAE of three components of IMF combination is bigger than S-MAE. When the input of our model includes Bx, By and Bz, it is

obviously that the MAE of poleward and equatorward boundaries both have significant increasement compared with the MAE of poleward and equatorward boundaries by inputting anyone IMF components to our model, which suggests that the three components of IMF have similar influence on auroral oval boundary. Previous investigations illustrated that auroral oval boundary is connected with the variation of IMF Bx, By, Bz (Huang et al., 2000; Provan et al., 1999). Our experiment results also demonstrate that the auroral oval boundary should be related with the three components of IMF.

We can see that AE has strong positively correlation to AU and PC, and AL has strong negative correlation to AE, AU and PC from Table 4. The linear correlation coefficient between AE and AL is -0.9437, which verified that AL index has the opposite effect on auroral boundary compared with AE index. In contrast, the impact of AU index on auroral boundary is similar to the impact of AE index on auroral boundary because of the strong positive correlation be-tween AE and AU. Those conclusions mentioned above is consistent with the conclusions of the statistical experiment 3. In addition, the correlation

coefficient between AE and PC is 0.8067, which implies PC should have homologous trend with AE in every MLT section. Fig.8 shows the response of poleward and equatorward boundaries to PC respectively. The PC index can serve as an indicator of auroral electrojet activity. Vennerstrøm found that PC is sensitive to electrojet activity and substorm intensifications of the westward electrojet in the midnight or post-midnight sector (Vennerstrøm et al., 1991). This conclusion matches to what we found about the impact of PC on auroral oval boundary in Fig.8. When the input of our model only included the three

geomagnetic indices (AE, AL, AU) and auroral oval boundary positions at the previous moment, the MAE of equatorward and poleward boundaries are 1.6569 and 1.5124 severally from Table 5. we can clearly know that the MAE of poleward and equatorward boundaries are both enlarged compared with S-MAE. As a result, we can draw the conclusion that the three geomagnetic indexes strengthen each other's effect when the combination of three geomagnetic indexes are inputted into our model. Beyond that, in Table 5, when AE, AU, AL, and PC are used as the input of our model, the MAE of poleward and

equatorward boundaries are 1.6734 and 1.5275 respectively, which shows that the combinations of those parameters have the important in-fluence on the location of aurora oval boundary.

**Table 5: Pearson correlation coefficient of some space physical parameters from Dec. 1996 to Mar. 1997**

| Parameter name | MAE(poleward/equatorward) |
|---|---|
| IMF | 1.6313/1.4759 |
| Solar wind index | 1.6495/1.4877 |
| Geomagnetic index | 1.6569/1.5124 |
| AE, AU, AL, PC | 1.6734/1.5272 |
| Vp, Np, Pdyn, SYM/H | 1.6611/1.4919 |

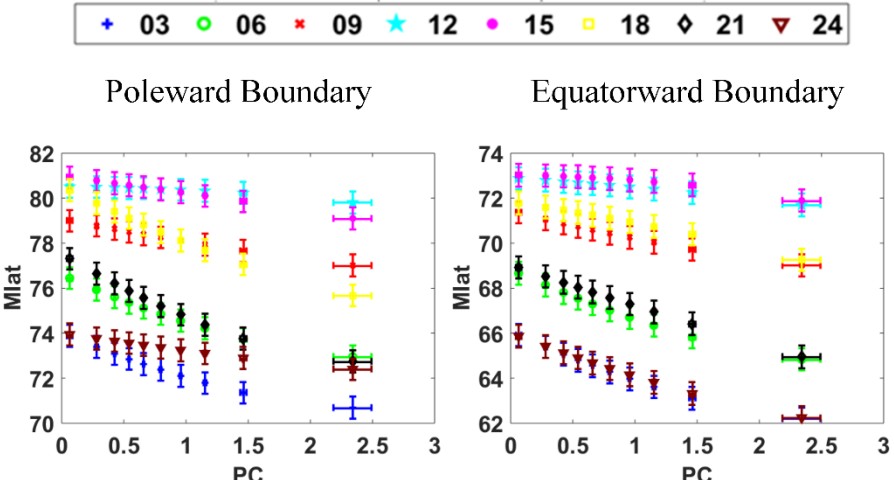

**Figure 8: Response of magnetic latitude of poleward (left column) and equa-torward (right column) boundaries to PC at 0030, 0060, 0090, 1200, 1500,1800,2100 and 2400 MLT**

According to Table 4, there has obvious correlation among the following space physical parameters. For the solar wind
parameters, Vp and Np are positive correlations, while Np and Pdyn are negative correlation, and the three parameters all are
related to SYM/H. Firstly, we can obtain the similar inference on the three geomagnetic indices to solar wind parameters (Vp,
Np, Pdyn) according to the MAE of three solar wind parameter combinations from Table 5 and the strong correlation between
them. In other words, the three solar wind parameters also strengthen each other's effect on auroral oval boundary when the
combination of them are sent into our model. Secondly, from Table 5, when Np, Vp, Pydn and SYM/H are sent to our model,
the MAE of auroral oval boundary reaches the minimum. They are 1.6611 and 1.4919 respectively, which shows that the
combinations of these four parameters have great influence on the location of aurora oval boundary.

As a summary, it can be seen that these space physical parameters shown in Table 5 play a crucial role in determining the
location of auroral oval boundary based on the above conclusions.

## 4 Conclusion

In this paper, we establish a model to measure the relation-ship between space physical parameters from OMNI dataset on
NASA website and poleward and equatorward auroral oval boundaries based on deep learning network. Our model overcomes
some drawbacks in this field. Such as, some prediction method based on statistics and a few space physical parameters. Those
method based is not very suitable for the complex and changeable space physical data. For our model, the inputs are 18 space
physical parameters and the 48 coordinates value of aurora oval boundary points at the previous moment, and we can obtain
position of poleward and equatorward boundaries at 24 MLTs from our well-trained model. At last, our experiment results
show that the model proposed in this paper can better reflect the relationship between space physical parameters and auroral
oval boundary. Therefore, it should be useful to predict the position of auroral oval boundary. In addition, we analyze the effect



of all 18 space physical parameters on the location of auroral oval boundary based on several statistical and prediction experiments. It can be show that different parameters have different effect on auroral oval boundary from our experiments. Some space physical parameters have a great influence on the position of auroral oval boundary, especially the space physical parameters which are shown in Table 5.

*Author contributions*. The methodology has been developed by all of the authors. HY. have coded and run the experiments. All authors have discussed the theory, the interpretation of the results and edited the manuscript.

*Competing interests*. The authors declare that they have no conflict of interest.

*Acknowledgements.* The authors are thankful to Liu Jianjun for his comments and suggestions. HB have been funded by the National Natural Science Foundation of China (61572384),Shaanxi Key Technologies Research Program (2017KW-017), China's Postdoctoral Fund First-class Funding (2014M560752), Shaanxi Province Postdoctoral Science Fund, the central
20   university basic scientific research business fee (JBG150225). HZ. have been funded by the National Natural Science Foundation of China (41874195). GX. have been funded by the National Key Research and Development Program of China (2016QY01W0200). YH. have been funded by the National Natural Science Foundation of China (41831072). LB have been funded by the National Natural Science Foundation of China (41504116). LJ. have been funded by the National Natural Science Foundation of China (41674169) and the National Key Research and Development Program of China
25   (2018YFC1407300).



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
