# Peer review of "Prediction and variation of auroral oval boundary based on deep learning model and space physical parameters"

_Nonlinear Processes in Geophysics, 2019_

## Referee Comment (RC1) · Anonymous Referee #1 · 2 Jul 2019

The reviewed manuscript contains some interesting statistical findings based on advanced machine learning methods.

My main concern is that the mathematical correlations between the position and size of the auroral oval and some physical variable describing the near-Earth space environment are not interpreted in physical terms. It's been known for a long time that the aurorally oval represents the open/closed field boundary of the Earth's magnetosphere which is responsive to the solar wind driver and reflects the dynamics of the Dungey substorm cycle. A careful discussion of these physical processes in the context of the reported machine learning results is required before this manuscript can be considered

for publication. What do the machine learning correlations show ? Do they support or challenge the existing substorm models? Do the enable a more accurate prediction of substorm magnitude and timing ? To make this analysis more informative it would be important to differentiate between the southward and northward IMF directions associated with drastically different solar wind driving conditions.

The presentation style is clear but there are multiply typing errors; I encourage the authors to proofread their manuscript before resubmitting.

---

## Author Comment (AC1) · 27 Jul 2019

Dear Reviewer, Thank you for your thorough and very constructive review. we reply to your main points and write a detailed reply in this document.

The main purpose of this paper is to explore the possible relationship between physical parameters and auroral events in a new way. In our previous work, some useful machine learning methods, such as deep learning and generalized regression neural network (GRNN) are used to model the relationship between the IMF and solar wind indexes and the auroral oval boundaries[1]. But we have no certain fact that whether the physical variables except IMF and solar wind indexes can affect auroral oval bound-

ary. Therefore, we want to obtain some unknown conclusions between more physical variables and auroral oval boundary by using our method in this paper.So, this paper is devoted to designing a new method rather than exploring the specific physical explanations between physical parameters and auroral oval boundary. The more exact physical connections for specific aurora events will be analyzed in future works.

In the manuscript, we design a new nonlinear and trainable mapping model to construct the relationship between auroral oval boundary and complex space physical parameters (including 18 parameters) to discuss the influence of every single space physical parameter on auroral oval boundary sufficiently. The experiment results in our paper show that some space physical parameters play a crucial role to locate the auroral oval position, which is consistent with the conclusion available in most of references. In the other hand, some interesting results have been obtained during the experiment. These conclusions maybe discuss with space physics in the future. In the task of predicting the substorm magnitude and timing, we cannot give an exact conclusion now. Our proposed method can model the relationship between different types of data, such as space physical data and auroral oval images. So, we think the proposed model in this manuscript can predict the substorm magnitude and timing after adjusting the model parameters according to the occurrence conditions and the data forms of substorm. We can give some researches on this problem in the future work. As a summary, our proposed model in this paper give the new way to analysis the relationship between many space physical parameters (such as: PC and SYM/H) which do not mentioned in other literature sand auroral oval boundary.

According to the reviewer comments, the additional two experiments are performed under southward and northward IMF directions conditions respectively according to your suggestion. The experiment results are shown in Table 5(Figure 1 can be considered as Table 5 in this reply document). Whether the northward or southward IMF direction are input to the proposed model, the MAE have marked change in both poleward and equatorward boundaries. Nevertheless, we can observe the more evident

increasement of MAE in equatorward boundaries compared with the MAE of poleward boundaries by using northward IMF direction as the input of our model. Meanwhile, there has an opposite result under southward IMF direction condition. The variation of MAE in poleward boundaries are bigger than equatorward boundaries when the input of our model is southward IMF direction. Therefore, we can know that the northward IMF direction has a great influence on the equatorward boundaries, and the southward IMF direction has a significant effect on poleward boundaries.

According to your comments, We already carefully check our manuscript to correct the typing errors. Details are given below.

(1) Line 3 in Introduction, the sentence 'which can implicit for the coupling process between solar wind, ionosphere and magnetosphere'. was revised to 'which can implicit for the coupling process among solar wind, ionosphere and magnetosphere'.

(2) Line 5 in Introduction,the sentence 'So, the segmentation and prediction for auroral oval boundary is very significant for studying on certain physical events'. was revised to 'So, the segmentation and prediction for auroral oval boundary are very significant for studying on certain physical events'.

(3) Line 12 in Introduction, the sentence 'Variations of the size of polar cap, auroral oval and diffuse aurora are regarded as three independent function variables of AL index (Starkov, 1994(b)).'was revised to 'Variations of the size of polar cap, auroral oval and diffuse aurora were regarded as three independent function variables of AL index respectively (Starkov, 1994(b))'.

(4) Line 15 in Introduction,the sentence 'Sigernes compared methods which proposed by Zhang and Starkov to calculate the size and position of auroral oval using a Kp-based function.' was revised to 'Sigernes used a Kp-based function to calculate the size and position of auroral oval, and compared the Kp-dependent model with methods which proposed by Zhang and Starkov to explain the superiority of his proposed model.'

(5) Line 41 in Introduction, the sentence 'Sect. 2 describes our proposed algorithm to predict auroral oval boundary in detail.' was revised to 'Sect. 2 describes our proposed algorithm in detail.'

(6) Line 3 in section 2, the sentence 'In the training phase, auroral oval images are usually affected by heavy noise and other interference.' was revised to 'In the training phase, auroral oval images are usually affected by heavy noise and other interferences.'

(7) Line 15 in section 2, the sentence 'The computational processing of RBM and RBF is illustrated by Eq. (1)-(4).' was revised to 'The computational processing of RBM and RBF are illustrated by Eq. (1)-(4).'

(8) Line 14 in section 3.1, the sentence 'According to the effect from other circumstance factors,' was revised to 'According to the effect derived from other circumstance factors,'

(9) Line 22 in section 3.1, the sentence 'Table 1 shows 18 space physical parameters we used in this paper.' was revised to 'Table 1 shows 18 space physical parameters which we used in this paper.' (10) Line 7 in section 3.2, 'The corresponding MAE is shown in Fig. 3(a).' was revised to 'The corresponding MAE are shown in Fig. 3(a).'

(11) Line 12 in section 3.2, the sentence 'The corresponding MAE is shown in Fig. 3(b), and MAE reaches the minimum when the training error of RBF net-work is 4 magnetic latitudes.' was revised to 'The corresponding MAE are shown in Fig. 3(b), and MAE reaches the minimum when the training error of RBF network is 4 magnetic latitudes.'

(12) Line 16 in section 3.2, the sentence 'To demonstrate the availability of our proposed model, we compare the proposed model with Back Propagation (BP) network (Rumelhart, 1986) and Yang's model (Yang et al., 2016).' was revised to 'To demonstrate the availability of our proposed model, we compared the proposed model with Back Propagation (BP) network (Rumelhart, 1986) and Yang's model (Yang et al., 2016).'

(13) Line 17 in section 3.2, the sentence 'The subjective prediction results obtained by

the three method are shown in Fig. 4, circles and squares stand for poleward boundary points and equatorward boundary points which are obtained from the segmented image, 'ïijŃ' and '×' marks represent poleward boundary points and equatorward boundary points which are obtained from our prediction model.' was revised to 'The subjective prediction results obtained by the three method are shown in Fig. 4, circles and squares stand for poleward boundary points and equatorward boundary points respectively which are obtained from the segmented image, 'ïijŃ' and '×' marks represent poleward boundary points and equatorward boundary points respectively which are obtained from our prediction model.'

(14) Line 21 in section 3.2, the sentence 'it is obviously that our method can obtain the more accurate boundaries than the other two compared methods, where marked by blue rectangle and red rectangle in Figure 4.' was revised to 'it is obviously that our method can obtain more accurate boundaries than the other two compared methods, where marked by blue rectangle and red rectangle in Fig. 4.'

(15) Line 6 in section 3.3, the sentence 'we sort boundary data with respect to the value of all space physical parameters, and divide evenly boundary data into 10 groups.' was revised to 'we sort boundary data with respect to the value of all space physical parameters, and divide boundary data into 10 groups evenly.'

(16) Line 15 in section 3.3, the sentence 'And a correlation analysis experiment is con-structed to study the connection between combination of different space physical parameters and auroral oval boundary.' was revised to 'And a correlation analysis experiment is constructed to study the connection between combination of different space physical parameters and auroral oval boundary.'

(17) Line 16 in section 3.3.1, the sentence 'Such as, Karlson's observations suggest-sthat IMF By component is related to prenoon-postnoon asymmetry of poleward activity (Karlson et al., 1996).' was revised to 'Such as, Karlson's observations suggested that IMF By component is related to prenoon-postnoon asymmetry of poleward activity

(Karlson et al., 1996).'

(18) Line 11 in section 3.3.4, the sentence 'The MAE of boundary position is 1.6076 and 1.4545 respectively when we only use boundary positions at the previous moment to predict poleward and equatorward boundaries. We take this MAE as standard, called S-MAE.' was revised to 'The MAE of boundary position are 1.6076 and 1.4545 respectively when we only use boundary positions at the previous moment to predict poleward and equatorward boundaries. We take this MAE as standard, called S-MAE.'

(19) Line 1 in section 3.3.5, the sentence 'For the sake of analyzing the influence of space physical parameters on auroral oval efficiently,' was revised to 'In order to analyse the influence of space physical parameters on auroral oval efficiently,'

(20) Line 7 in section 3.3.5, the sentence 'The formula of Pearson correlation coefficient is can presented as Eq. (6),' was revised to 'The formula of Pearson correlation coefficient can presented as Eq. (6),'

(21) Line 4 in Conclusion, the sentence 'Those method based is not very suitable for the complex and changeable space physical data.' was revised to 'Those methods are not very suitable for the complex and changeable space physical data.'

(22) Line 9 in Conclusion, the sentence 'we analyze the effect of all 18 space physical parameters on the location of auroral oval boundary based on several statistical and prediction experiments.' was revised to 'we analyse the effect of all 18 space physical parameters on the location of auroral oval boundary based on several statistical and prediction experiments.'
* * *
**Table 5:** Pearson correlation coefficient of some space physical parameters from Dec. 1996 to Mar. 1997

| Parameter name | MAE(poleward/equatorward) | |
|---|---|---|
| IMF | 1.6313/1.4759 | |
| IMF(Bz>0) | 1.6365/1.7595 | |
| IMF(Bz<0) | 1.7163/1.6193 | |
| Solar wind index | 1.6495/1.4877 | |
| Geomagnetic index | 1.6569/1.5124 | |
| AE, AU, AL, PC | 1.6734/1.5272 | |
| Vp, Np, Pdyn, SYM/H | 1.6611/1.4919 | |

**Fig. 1.**

---

## Short Comment (SC1) · 5 Aug 2019

Review of the manuscript "Prediction and variation of auroral oval boundary based on deep learning model and space physical parameters" byYiyuan Han et al.

This manuscript describes an automatic auroral oval boundary prediction method by using a new deep learning model and shows some interesting results about the relationship between the location of auroral oval boundaries and several space physical parameters. A number of experiments are performed to test the proposed model and the connections between different physical parameters and the acquired auroral oval boundaries. This manuscript gives some new insights into the prediction of auroral oval

boundaries variation.

Some comments and suggestions: 1. This manuscript designed a new model based on deep learning to construct the relationship between the physical variables and auroral oval boundaries. Did the authors consider some specific aurora forms (such as, substorms,transpolar arcs) when you did the experiments? Is it possible to construct the connection among other forms of data? Such as: the relationship between physical variables and auroral oval intensity in the process of a specific aurora event (for example: substorms, polar cap arc). Please give an expatiation.

2. As we known that the aurora dynamics are influenced by many factors, not only the variation of physical variables. Why the authors chose these 18 space physical parameters? Could you give an explanation? Whether this model can establish a connection between a specific aurora event and multiple forms of data, not only one parameter of data?

3. In the Introduction section, the authors described many previous studies. It seems to lack intrinsic and progressive connections among these previous studies. Could you rewrite them and make them more logically?

4. In section 3.3, please explain why the authors use the boundary data which calculated by 'Quadratic Equation' instead of the original boundary data to discuss the influence of space physical parameters on auroral oval boundary.

5. Some figures in this manuscript are not clear enough, and the readers may need high quality figures.

6. When an abbreviation of term first appears in the manuscript, the term should have a full name first. Such as MRSM, MLT-MLAT et al. Please check them.

7. The paper is easy to follow, but there are some writing mistakes or misleading descriptions which make the readers confused. Such as, Page 1, Line 24 "ex-tensive"; Page 2, Line 14 "be-tween"; Line 22, 26. Figure caption of Figure 5-7, should not be

"left column", "right column". Please check and revise them. Please proofread this manuscript carefully.

I hope the authors recheck them and can take the above comments into account.

Please also note the supplement to this comment: https://www.nonlin-processes-geophys-discuss.net/npg-2019-28/npg-2019-28-SC1-supplement.pdf

---

## Referee Comment (RC2) · Unnikrishnan Kaleekkal (Referee) · 21 Sep 2019

Review of manuscript, "Prediction and variation of auroral oval boundary based on deep learning model and space physical parameters", by Yiyuan Han et al.,

In this manuscript, authors have proposed a new automatic auroral oval boundary prediction model, based on deep learning method, using space physical parameters, and the location of auroral oval boundary at the previous moment. The proposed model is well explained with flow chart, and the procedure of training/testing of sufficient data sets supported by interpretation of results. A significant aspect of this model is, several probable parameters that can influence the variation of auroral oval have been

fed as the inputs, which is a prerequisite to model complex system. More specifically, 18 space physical parameters and the 48 coordinates value of aurora oval boundary points at the previous moment were utilised for training/ testing, leading to the reasonable prediction of the position of poleward and equatorward boundaries at 24 MLTs. Identification of optimum choice of input parameters is a crucial aspect. In this work, it is shown that, different space physical parameters have different effects on auroral oval boundary, especially interplanetary magnetic field (IMF), geomagnetic indexes and solar wind parameters. Out of the input parameters used for training and testing the present model, authors can check how does the prediction capability of the model vary when inputs with least significance were removed. Or they can add some more probable parameters as inputs and observe the performance of the model. Finally, a combination of input parameters can be selected, based on which model performance is highest. This kind of a procedure will help to identify an optimum choice of input parameters, thereby establishing an input-output relation, and refine the existing model further. The authenticity of the proposed model can be improved further, by this way. Generally, the manuscript is well written, and the proposed model exhibits good prediction capabilities by delivering interesting results.

---

## Author Comment (AC3) · 28 Sep 2019

We are very grateful and convey our sincere thank you to your generous comments and significant suggestion for the manuscript. We have revised the manuscript to best of our knowledge as per your comments and suggestions. We have submitted our response in the "npg-2019-28-RC1-supplement.zip" folder as a supplement which contains pdf documents of the following: 1. A SoC document to Anonymous Referee #1, named 'npg-2019-28-SoC_RC1'. 2. A revised manuscript according the comments of all reviewers, named 'R1_Prediction and variation of auroral oval boundary based on deep learning model and space physical parameters.'

Please also note the supplement to this comment:
https://www.nonlin-processes-geophys-discuss.net/npg-2019-28/npg-2019-28-AC3-supplement.zip

―――――――――――――――――

---

## Author Comment (AC4) · 28 Sep 2019

We are very grateful and convey our sincere thank you to your generous comments and significant suggestion for the manuscript. We have revised the manuscript to best of our knowledge as per your comments and suggestions. We have submitted our response in the "npg-2019-28-SC-supplement.zip" folder as a supplement which contains pdf documents of the following: 1. A SoC document to reviewer, named 'npg-2019-28-SoC_SC'. 2. A revised manuscript according the comments of all reviewers, named 'R1_Prediction and variation of auroral oval boundary based on deep learning model and space physical parameters.'

Please also note the supplement to this comment:
https://www.nonlin-processes-geophys-discuss.net/npg-2019-28/npg-2019-28-AC4-supplement.zip

───────────────────────

---

## Author Comment (AC5) · 28 Sep 2019

We are very grateful and convey our sincere thank you to your generous comments and significant suggestion for the manuscript. We have revised the manuscript to best of our knowledge as per your comments and suggestions. We have submitted our response in the "npg-2019-28-RC2-supplement.zip" folder as a supplement which contains pdf documents of the following: 1. A SoC document to reviewer, named 'npg-2019-28-SoC_RC2'. 2. A revised manuscript according the comments of all reviewers, named 'R1_Prediction and variation of auroral oval boundary based on deep learning model and space physical parameters.'

Please also note the supplement to this comment:
https://www.nonlin-processes-geophys-discuss.net/npg-2019-28/npg-2019-28-AC5-supplement.zip

---

## Author Comment (AC6) · 28 Sep 2019

Thanks for the kind suggestion. We have conducted the experiments to find how will the prediction capability of the proposed model change when the inputs are different combinations of physical variables. And we considered all the possible combinations of physical variables which we mentioned in Table 5 to find an optimum choice of input parameters. However, we cannot give all the experiment results in this manuscript because of the huge numbers of all supplement experiments. Therefore, we give the best experiment results (MAE value) obtained from different combinations of physical variables in Table 6. And other results are given in the supplementary material, which

is a Excel file, named 'The experiment results of Optimal input search'.

Please also note the supplement to this comment:
https://www.nonlin-processes-geophys-discuss.net/npg-2019-28/npg-2019-28-AC6-supplement.zip

---

## Author Response (AR2)

**Summary of Changes**

The authors would like to thank all the reviewers for their constructive comments. We provide below a detailed account on the changes that we have made in response to the comments that the editor and the reviewers have raised. For comments from different reviewers, we have marked the corresponding changes in the revised version in *red, green and purple* color respectively.

**Response to Anonymous Referee #1:**

The reviewed manuscript contains some interesting statistical findings based on advanced machine learning methods.

1. My main concern is that the mathematical correlations between the position and size of the auroral oval and some physical variable describing the near-Earth space environment are not interpreted in physical terms. It's been known for a long time that the aurorally oval represents the open/closed field boundary of the Earth's magnetosphere which is responsive to the solar wind driver and reflects the dynamics of the Dungey substorm cycle. A careful discussion of these physical processes in the context of the reported machine learning results is required before this manuscript can be considered for publication.

*Author's reply:*

The main purpose of this paper is to explore the possible relationship between physical parameters and auroral events in a new way. In our previous work, some useful machine learning methods, such as deep learning and generalized regression neural network (GRNN) are used to model the relationship between the IMF and solar wind indexes and the auroral oval boundaries[1]. But we have no certain fact that whether the physical variables except IMF and solar wind indexes can affect auroral oval boundary. Therefore, we want to obtain some unknown conclusions between more physical variables and auroral oval boundary by using our method in this paper. So, this paper is devoted to designing a new method rather than exploring the specific physical explanations between physical parameters and auroral oval boundary. The more exact physical connections for specific aurora events will be analyzed in future works.

[1] HanB, LianHF, HuZJ. Modeling of ultraviolet auroral oval boundaries based on neural network technology (inChinese). SciSinTech, 2019, 49:531–542, doi: 10.1360/N092018-00227.

2. What do the machine learning correlations show? Do they support or challenge the existing substorm models? Do the enable a more accurate prediction of substorm magnitude and timing?

*Author's reply:*

In the manuscript, we design a new nonlinear and trainable mapping model to construct the relationship between auroral oval boundary and complex space physical parameters (including 18 parameters) to discuss the influence of every single space physical parameter on auroral oval boundary sufficiently. The experiment results in our paper show that some space physical parameters play a crucial role to locate the auroral oval position, which is consistent with the conclusion available in most of references. In the other hand, some interesting results have been obtained during the experiment. These conclusions maybe discuss with space physics in the future.

In the task of predicting the substorm magnitude and timing, we cannot give an exact conclusion now. Our proposed method can model the relationship between different types of data, such as space physical data and auroral oval images. So, we think the proposed model in this manuscript can predict the substorm magnitude and timing after adjusting the model parameters according to the occurrence conditions and the data forms of substorm. We can give some researches on this problem in the future work. As a summary, our proposed model in this paper give the new way to analysis the relationship between many space physical parameters (such as: PC and SYM/H) which not mentioned in other literature sand auroral oval boundary.

3. To make this analysis more informative it would be important to differentiate between the southward and northward IMF directions associated with drastically different solar wind driving conditions.

*Author's reply:*

According to the reviewer comments, the additional two experiments are performed under southward and northward IMF directions conditions respectively according to your suggestion. The experiment results are shown in Table 5.

*Revision:*

**Table 5: Pearson correlation coefficient of some space physical parameters from Dec. 1996 to Mar. 1997**

| Parameter name | MAE(poleward/equatorward) |
|---|---|
| IMF | 1.6313/1.4759 |
| Solar wind index | 1.6495/1.4877 |
| Geomagnetic index | 1.6569/1.5124 |
| AE, AU, AL, PC | 1.6734/1.5272 |
| Vp, Np, Pdyn, SYM/H | 1.6611/1.4919 |

[Figure]

**Table 5: The MAE influenced by different combinations of space physical parameters.**

| Parameter name | MAE(poleward/equatorward) |
|---|---|
| IMF | 1.6313/1.4759 |
| IMF(Bz>0) | 1.6365/1.7595 |
| IMF(Bz<0) | 1.7163/1.6193 |
| Solar wind index | 1.6495/1.4877 |
| Geomagnetic index | 1.6569/1.5124 |
| AE, AU, AL, PC | 1.6734/1.5272 |
| Vp, Np, Pdyn, SYM/H | 1.6611/1.4919 |

Whether the northward or southward IMF direction are input to the proposed model, the MAE have marked change in both poleward and equatorward boundaries. Nevertheless, we can observe the more evident increasement of MAE in equatorward boundaries compared with the MAE of poleward boundaries by using northward IMF direction as the input of

our model. Meanwhile, there has an opposite result under southward IMF direction condition. The variation of MAE in poleward boundaries are bigger than equatorward boundaries when the input of our model is southward IMF direction. Therefore, we can know that the northward IMF direction has a great influence on the equatorward boundaries, and the southward IMF direction has a significant effect on poleward boundaries.

4. The presentation style is clear but there are multiply typing errors; I encourage the authors to proofread their manuscript before resubmitting.

***Author's reply:***

Thanks for your kind suggestion. We already carefully check our manuscript to correct the typing errors. Details are given below.

***Revision:***

(1) Line 3 in Introduction, 'which can implicit for the coupling process between solar wind, ionosphere and magnetosphere'. → 'which can implicit for the coupling process among solar wind, ionosphere and magnetosphere'.

(2) Line 5 in Introduction, 'So, the segmentation and prediction for auroral oval boundary is very significant for studying on certain physical events'. →'So, the segmentation and prediction for auroral oval boundary are very significant for studying on certain physical events'.

(3) Line 12 in Introduction, 'Variations of the size of polar cap, auroral oval and diffuse aurora are regarded as three independent function variables of AL index (Starkov, 1994(b)).' → 'Variations of the size of polar cap, auroral oval and diffuse aurora were regarded as three independent function variables of AL index respectively (Starkov, 1994(b))'.

(4) Line 15 in Introduction, 'Sigernes com-pared methods which proposed by Zhang and Starkov to calculate the size and position of auroral oval using a Kp-based function.' →'Sigernes used a Kp-based function to calculate the size and position of auroral oval, and compared the Kp-dependent model with methods which proposed by Zhang and Starkov to explain the superiority of his proposed model.'

(5) Line 41 in Introduction, 'Sect. 2 describes our proposed algorithm to predict auroral oval boundary in detail.' →'Sect. 2 describes our proposed algorithm in detail.'

(6) Line 3 in section 2, 'In the training phase, auroral oval images are usually affected by heavy noise and other interference.' →'In the training phase, auroral oval images are usually affected by heavy noise and other interferences.'

(7) Line 15 in section 2, 'The computational processing of RBM and RBF is illustrated by Eq. (1)-(4).' →'The computational processing of RBM and RBF are illustrated by Eq. (1)-(4).'

(8) Line 14 in section 3.1, 'According to the effect from other circumstance factors,' →'According to the effect derived from other circumstance factors,'

(9) Line 22 in section 3.1, 'Table 1 shows 18 space physical parameters we used in this paper.' →'Table 1 shows 18 space physical parameters which we used in this paper.'

(10) Line 7 in section 3.2, 'The corresponding MAE is shown in Fig. 3(a).' →'The corresponding MAE are shown in Fig. 3(a).'

(11) Line 12 in section 3.2, 'The corresponding MAE is shown in Fig. 3(b), and MAE reaches the minimum when the training error of RBF net-work is 4 magnetic latitudes.' → 'The corresponding MAE are shown in Fig. 3(b), and MAE reaches the minimum when the training error of RBF network is 4 magnetic latitudes.'

(12) Line 16 in section 3.2, 'To demonstrate the availability of our proposed model, we compare the proposed model with Back Propagation (BP) network (Rumelhart, 1986) and Yang's model (Yang et al., 2016).' → 'To demonstrate the availability of our proposed model, we compared the proposed model with Back Propagation (BP) network (Rumelhart, 1986) and Yang's model (Yang et al., 2016).'

(13) Line 17 in section 3.2, 'The subjective prediction results obtained by the three method are shown in Fig. 4, circles and squares stand for poleward boundary points and equatorward boundary points which are obtained from the segmented image, '＋' and '×' marks represent poleward boundary points and equatorward boundary points which are obtained from our prediction model.' → 'The subjective prediction results obtained by the three method are shown in Fig. 4, circles and squares stand for poleward boundary points and equatorward boundary points respectively which are obtained from the segmented image, '＋' and '×' marks represent poleward boundary points and equatorward boundary points respectively which are obtained from our prediction model.'

(14) Line 21 in section 3.2, 'it is obviously that our method can obtain the more accurate boundaries than the other two compared methods, where marked by blue rectangle and red rectangle in Figure 4.' → 'it is obviously that our method can obtain more accurate boundaries than the other two compared methods, where marked by blue rectangle and red rectangle in Fig. 4.'

(15) Line 6 in section 3.3, 'we sort boundary data with respect to the value of all space physical parameters, and divide evenly boundary data into 10 groups.' → 'we sort boundary data with respect to the value of all space physical parameters, and divide boundary data into 10 groups evenly.'

(16) Line 15 in section 3.3, 'And a correlation analysis experiment is con-structed to study the connection between combination of different space physical parameters and auroral oval boundary.' → 'And a correlation analysis experiment is constructed to study the connection between combination of different space physical parameters and auroral oval boundary.'

(17) Line 16 in section 3.3.1, 'Such as, Karlson's observations suggests that IMF By component is related to prenoon-postnoon asymmetry of poleward activity (Karlson et al., 1996).' → 'Such as, Karlson's observations suggested that IMF By component is related to prenoon-postnoon asymmetry of poleward activity (Karlson et al., 1996).'

(18) Line 11 in section 3.3.4, 'The MAE of boundary position is 1.6076 and 1.4545 respectively when we only use boundary positions at the previous moment to predict poleward and equatorward boundaries. We take this MAE as standard, called S-MAE.' → 'The MAE of boundary position are 1.6076 and 1.4545 respectively when we only use boundary positions at the previous moment to predict poleward and equatorward boundaries. We take this MAE as standard, called S-MAE.'

(19) Line 1 in section 3.3.5, 'For the sake of analyzing the influence of space physical parameters on auroral oval efficiently,' → 'In order to analyse the influence of space physical parameters on auroral oval efficiently,'

(20) Line 7 in section 3.3.5, 'The formula of Pearson correlation coefficient is can presented as Eq. (6),' → 'The formula of

Pearson correlation coefficient can be represented as Eq. (6),'

(21) Line 4 in Conclusion, 'Those method based is not very suitable for the complex and changeable space physical data.' → 'Those methods are not very suitable for the complex and changeable space physical data.'

(22) Line 9 in Conclusion, 'we analyze the effect of all 18 space physical parameters on the location of auroral oval boundary based on several statistical and prediction experiments.' → 'we analyse the effect of all 18 space physical parameters on the location of auroral oval boundary based on several statistical and prediction experiments.'

**Response to Reviewer Zanyang Xing:**

1. This manuscript designed a new model based on deep learning to construct the relationship between the physical variables and auroral oval boundaries. Did the authors consider some specific aurora forms (such as, substorms, transpolar arcs) when you did the experiments? Is it possible to construct the connection among other forms of data? Such as: the relationship between physical variables and auroral oval intensity in the process of a specific aurora event (for example: substorms, polar cap arc). Please give an expatiation.

*Author's reply:*

In this paper, the main purpose is only to explore the possible and unknown relationship between physical parameters and auroral oval boundary over a period of time. But it is possible to occur any auroral events during December 1996 to March 1997. We analyzed all phenomenon that we observed in our experiments(Section 3.3.1-3.3.5),more than the relationship between specific aurora event (for example: substorms) and the variations of some specific physical parameters (such as, AE).The advantage of our model is that it can construct the connection among different forms of data. So, we think the proposed model in this manuscript can predict the auroral oval intensity by adjusting the model parameters according to the occurrence conditions and the data forms of substorm or polar cap arc. We can give some researches on this problem the reviewer mentioned in the future work.

2. As we known that the aurora dynamics are influenced by many factors, not only the variation of physical variables. Why the authors chose these 18 space physical parameters? Could you give an explanation? Whether this model can establish a connection between a specific aurora event and multiple forms of data, not only one parameter of data?

*Author's reply:*

In this paper, we chose 18 space physical parameters from OMNI dataset to train our model. Firstly, the common physical variables include IMF, solar wind parameters and geomagnetic indexes, which were discussed in previous work (Niu et al., 2015; Milan, 2010; Hu et al., 2017; Yang et al., 2016). The remaining physical variables are related to those common physical variables. So, we select these 18 space physical parameters as the research targets in this manuscript.

There are two ways to use multiple forms of data as the input of our model. Firstly, we can unify the different forms of data into a same data space. The other way is that we can extend our model as a new model which can input multi-source data. These two methods both can address the problems of establishing a relationship between multiple forms of data and a specific aurora event.

3. In the Introduction section, the authors described many previous studies. It seems to lack intrinsic and progressive connections among these previous studies. Could you rewrite them and make them more logically?

*Author's reply:*

In the Introduction section, we revised the part of previous studies according your suggestion. Details are as follow.

*Revision:*

(1) Page 1, Line 25, In early research, Feldstein proposed that the position of auroral oval boundary is correlated with the Q-

index of magnetic activity on the nightside of earth (Feldstein and Starkov, 1967). Starkov and Holzworth expressed that inner and outer boundaries of auroral oval can change with geomagnetic indexes and IMF (Holzworth and Meng, 1975; Holzworth and Meng, 1984; Starkov, 1994(a)).

(2) Page 1, Line 28, Starkov designed some simple formulas on the location of auroral oval and diffuse aurora. Variations of the size of polar cap, → "The conclusions in this paper are based on mathematical statistics. Therefore, Starkov designed some simple formulas to describe the relationships between the specific physical parameter and different type of aurora. Variations of the size of polar cap,"

(3) Page 2, Line 2, Since then, many scholars had been explored the connections between different physical parameters and auroral oval boundary or other auroral events.

(4) Page 2, Line 11, After 2010, there are more and more new methods to construct connection between the position of auroral oval boundary and auroral oval boundary with the development of machine learning.

4. In section 3.3, please explain why the authors use the boundary data which calculated by 'Quadratic Equation' instead of the original boundary data to discuss the influence of space physical parameters on auroral oval boundary.

*Author's reply:*

The original boundary data are chaotic because of the uncertain aurora events and complex space environment. It is hard to explore clearly variations of auroral oval boundary associated with space physical parameters. To obtain the variable patterns of auroral oval boundary clearly, we use the boundary data which calculated by 'Quadratic Equation' instead of the original boundary data in this paper.

5. Some figures in this manuscript are not clear enough, and the readers may need high quality figures.

*Author's reply:*

We use high quality figures instead of original figures in this manuscript according your suggestion. Details are as follow.

*Revision:*

[Figure]

Figure 5: Response of magnetic latitude of poleward (left column) and equatorward (right column) boundaries to Bx, By and Bz respectively at 0030,0060,0090,1200,1500,1800,2100 and 2400 MLT.

[Figure]

[Figure]

Figure 5: Response of magnetic latitude of poleward (top row) and equatorward (bottom row) boundaries to Bx, By and Bz respectively at 0030,0060,0090,1200,1500,1800,2100 and 2400 MLT.

6. When an abbreviation of term first appears in the manuscript, the term should have a full name first. Such as MRSM, MLT-MLAT et al. Please check them.

*Author's reply:*

We added the abbreviation of term first appears in the manuscript according your suggestion. Details are as follow.

*Revision:*

(1) In Abstract section, Line 5, "UVI image" → "Ultraviolet Imager (UVI) image";

(2) In Section 2, Line 4, "MRSM" → "Maximal Similarity Based Region Merging (MRSM)";

(3) In Section 2, Line 6, "magnetic local time-magnetic latitude coordinate" → "magnetic local time-magnetic latitude coordinate (MLT-MLAT)".

7. The paper is easy to follow, but there are some writing mistakes or misleading descriptions which make the readers confused. Such as, Page 1, Line 24 "ex-tensive"; Page 2, Line 14 "be-tween"; Line 22, 26. Figure caption of Figure 5-7, should not be "left column", "right column". Please check and revise them. Please proofread this manuscript carefully.

*Author's reply:*

We revised these writing mistakes and misleading descriptions according your suggestion. Details are as follow.

*Revision:*

(1) Page 1, Line 24, "In the past few decades, scholars have constructed ex-tensive researches on the relationship between location of auroral oval boundary and space physical parameters (Niu et al., 2015)"; → "In the past few decades, scholars

have constructed extensive researches on the relationship between location of auroral oval boundary and space physical parameters (Niu et al., 2015)";

(2) Page 2, Line 19, "those methods mentioned above just only used one or several space physical parameters to explore the relationship be-tween space physical parameters and auroral oval boundary." → "those methods mentioned above just only used one or several space physical parameters to explore the relationship between space physical parameters and auroral oval boundary."

(3) Page 2, Line 30, "The experiment results show that the model pro-posed in this paper can predict aurora oval boundary accurately by using space physical parameters and the location of auroral oval boundary at the previous moment." → "The experiment results show that the model proposed in this paper can predict aurora oval boundary accurately by using space physical parameters and the location of auroral oval boundary at the previous moment."

(4) Figure caption of Figure 5-7, "left column" → "top row", "right column" → "bottom row".

**Response to Reviewer Unnikrishnan Kaleekkal:**

In this manuscript, authors have proposed a new automatic auroral oval boundary prediction model, based on deep learning method, using space physical parameters, and the location of auroral oval boundary at the previous moment. The proposed model is well explained with flow chart, and the procedure of training/testing of sufficient data sets supported by interpretation of results.

1. A significant aspect of this model is, several probable parameters that can influence the variation of auroral oval have been fed as the inputs, which is a prerequisite to model complex system. More specifically, 18 space physical parameters and the 48 coordinates value of aurora oval boundary points at the previous moment were utilised for training/ testing, leading to the reasonable prediction of the position of poleward and equatorward boundaries at 24 MLTs. Identification of optimum choice of input parameters is a crucial aspect. In this work, it is shown that, different space physical parameters have different effects on auroral oval boundary, especially interplanetary magnetic field (IMF), geomagnetic indexes and solar wind parameters. Out of the input parameters used for training and testing the present model, authors can check how does the prediction capability of the model vary when inputs with least significance were removed. Or they can add some more probable parameters as inputs and observe the performance of the model. Finally, a combination of input parameters can be selected, based on which model performance is highest. This kind of a procedure will help to identify an optimum choice of input parameters, thereby establishing an input-output relation, and refine the existing model further. The authenticity of the proposed model can be improved further, by this way.

***Author's reply:***

Thanks for the kind suggestion. We have conducted the experiments to find how will the prediction capability of the proposed model change when the inputs are different combinations of physical variables. And we considered all the possible combinations of physical variables which we mentioned in Table 5 to find an optimum choice of input parameters. However, we cannot give all the experiment results in this manuscript because of the huge numbers of all supplement experiments. The number of experiments which can be calculated by the following equation:

$$C_{11}^2 + C_{11}^3 + C_{11}^4 + C_{11}^5 + C_{11}^6 + C_{11}^7 + C_{11}^8 + C_{11}^9 + C_{11}^{10} + C_{11}^{11} = 2036$$

Therefore, we give the best experiment results (MAE value) obtained from different combinations of physical variables in Table 6. And other results are given in the supplementary materials.

***Revision:***

1. According to the revision above described, we revised the manuscript further as followings

(1) In section 3.3.5:

[revised manuscript text omitted]